# Keep Various Trajectories: Promoting Exploration of Ensemble Policies in Continuous Control

**Chao Li**[1,4]    **Chen Gong**[2]    **Qiang He**[3]    **Xinwen Hou**[1†]
[1]Institute of Automation, Chinese Academy of Sciences, China
[2]University of Virginia, USA
[3]Ruhr University Bochum, Germany
[4]University of Chinese Academy of Sciences, China
{lichao2021, xinwen.hou}@ia.ac.cn
fzv6en@virginia.edu, qianghe97@gmail.com

## Abstract

The combination of deep reinforcement learning (DRL) with ensemble methods has been proved to be highly effective in addressing complex sequential decision-making problems. This success can be primarily attributed to the utilization of multiple models, which enhances both the robustness of the policy and the accuracy of value function estimation. However, there has been limited analysis of the empirical success of current ensemble RL methods thus far. Our new analysis reveals that the sample efficiency of previous ensemble DRL algorithms may be limited by sub-policies that are not as diverse as they could be. Motivated by these findings, our study introduces a new ensemble RL algorithm, termed **T**rajectories-awar**E E**nsemble exploratio**N** (TEEN). The primary goal of TEEN is to maximize the expected return while promoting more diverse trajectories. Through extensive experiments, we demonstrate that TEEN not only enhances the sample diversity of the ensemble policy compared to using sub-policies alone but also improves the performance over ensemble RL algorithms. On average, TEEN outperforms the baseline ensemble DRL algorithms by 41% in performance on the tested representative environments.

## 1   Introduction

Deep Reinforcement Learning (DRL) [1] has demonstrated significant potential in addressing a range of complex sequential decision-making problems, such as video games [2; 3; 4], autonomous driving [5; 6], robotic control tasks [7; 8], board games [9; 10], etc. Although the combination of high-capacity function approximators, such as deep neural networks (DNNs), enables DRL to solve more complex tasks, two notorious problems impede the widespread use of these DRL in real-world domains. i) *function approximation error:* Q-learning algorithm converges to sub-optimal solutions due to the error propagation [11; 12; 13], e.g., the maximization of random function approximation errors accumulate into consistent overestimation bias [14]. ii) *sample inefficiency*, model-free algorithm is notorious for requiring sample diversity—training of neural networks requires a large number of samples which is hard to acquire in real-world scenarios.

Ensemble reinforcement learning shows great potential in solving the issues mentioned above by combining multiple models of the value function and (or) policy [14; 15; 16; 17; 18; 19; 20] to improve the accuracy and diversity. For instance, Max-Min Q-learning [11] reduces estimation bias by utilizing an ensemble of estimates and selecting the minimum value as the target estimate. Bootstrapped DQN [15] and SUNRISE [16] train an ensemble of value functions and policies,

---

[†]Corresponding Author.

leveraging the uncertainty estimates of value functions to enrich the diversity of learning experiences. However, we claim that existing ensemble methods do not effectively address the crucial issue of diverse exploration, which is essential for enhancing sample efficiency. Although randomly initializing sub-policies shares some similarities to adding random noise with the ensemble policy, it is not sufficient for significantly improving the sample diversity of the ensemble policy. Thus, how to enhance the exploration diversity of ensemble policy remains an open question.

This paper presents a novel approach called **T**rajectories-awar**E E**nsemble exploratio**N** (TEEN) that encourages diverse exploration of ensemble policy by encouraging diverse behaviors through exploration of the state-action visit distribution measure space. The state-action visit distribution measure quantifies the frequency with which a particular state-action pair is visited when using a certain policy. Thus, policies with diverse state-action visit distributions induce different trajectories [21]. Moreover, value functions are learned from the trajectories generated by the policy, rather than the actions the policy may take. While previous research [22; 23; 24] has emphasized promoting diverse actions, this approach does not always lead to diverse trajectories. This paper highlights the importance of diverse trajectories in improving sample efficiency, which TEEN achieves.

Our contributions can be summarized in three aspects. (1) We propose **T**rajectories-awar**E E**nsemble exploratio**N** (TEEN) algorithm, a highly sample-efficient ensemble reinforcement learning algorithm. TEEN trains sub-policies to exhibit a greater diversity of behaviors, accomplished by considering the distribution of trajectories. (2) Theoretical analysis confirms that our algorithm facilitates more diverse exploration by enhancing the varied behaviors of the sub-policies. (3) Extensive experimental results show that our method outperforms or achieves a similar level of performance as the current state-of-the-art across multiple environments, which include both MuJoCo control [25] tasks and DMControl tasks [26]. Specifically, on average across the tasks we investigated, TEEN surpasses the baseline ensemble DRL algorithm by 41% and the state-of-the-art off-policy DRL algorithms by 7.3% in terms of performance. Furthermore, additional experiments demonstrate that our algorithm samples from a diverse array of trajectories.

## 2 Related Work

**Ensemble Methods.** Ensemble methods have been widely utilized in DRL to serve unique purposes. Overestimation bias in value function severely undermines the performance of Q-learning algorithms, and a variety of research endeavors on the ensemble method have been undertaken to alleviate this issue [14; 18; 19; 27; 11; 28]. TD3 [14] clips Double Q-learning to control the overestimation bias, which significantly improves the performance of DDPG [29]. Maxmin Q-learning [11] directly mitigates the overestimation bias by using a minimization over multiple action-value estimates. Ensemble methods also play a role in encouraging sample efficient exploration [30]. For example, Bootstrapped DQN [15] leverages uncertainty estimates for efficient (and deep) exploration. On top of that, REDQ [17] enhances the sample efficiency by using a update-to-data ratio $\gg 1$. SUNRISE [16] turns to an ensemble policy to consider for compelling exploration.

**Diverse Policy.** A range of works have been proposed for diverse policy, and these can generally be classified into three categories [31]. The first category involves directly optimizing policy diversity during pre-training without extrinsic rewards. Fine-tuning with rewards is then applied to complete the downstream tasks. These methods [32; 33; 34; 35] train the policy by maximizing the mutual information between the latent variable and the trajectories, resulting in multiple policies with diverse trajectories. The second category addresses constraint optimization problems, where either diversity is optimized subject to quality constraints, or the reverse is applied – quality is optimized subject to diversity constraints [36; 37; 38; 39]. The third category comprises quality diversity methods [31; 40; 41], which prioritize task-specific diversity instead of task-agnostic diversity. These methods utilize scalar functions related to trajectories and simultaneously optimize both quality and diversity.

**Efficient Exploration.** Efficient exploration methods share similar methods to diverse policy methods in training the agent, wherein exploration by diverse policies can enhance the diversity of experiences, ultimately promoting more efficient exploration. [42]. Other prior works focusing on efficient exploration consider diverse explorations within a single policy, with entropy-based methods being the representative approach [22; 43]. These methods strive to simultaneously maximize the expected cumulative reward and the entropy of the policy, thus facilitating diverse exploration. Additionally,

another representative category of research is the curiosity-driven method [24; 23; 44]. These methods encourage the agent to curiously explore the environment by quantifying the state with "novelty", such as state visit counts [23], model prediction error [24], etc.

# 3 Preliminaries

## 3.1 Markov Decision Process

Markov Decision Process (MDP) can be described as a tuple $\langle \mathcal{S}, \mathcal{A}, P, r, \gamma \rangle$, where $\mathcal{S}$ and $\mathcal{A}$ represent the state and action spaces, respectively. The function $P$ maps a state-action pair to the next state with a transition probability in the range $[0, 1]$, i.e., $P : \mathcal{S} \times \mathcal{A} \times \mathcal{S} \to [0, 1]$. The reward function $r : \mathcal{S} \times \mathcal{A} \to \mathbb{R}$ denotes the reward obtained by performing an action $a$ in a particular state $s$. Additionally, $\gamma \in (0, 1)$ is the discount factor. The goal of reinforcement learning algorithms is to find a policy $\pi^*$ that maximizes the discounted accumulated reward $J(\pi)$, defined as:

$$J(\pi) = \mathbb{E}_{s,a \sim \pi}[\Sigma_{t=0}^{\infty} \gamma^t r(s_t, a_t)] \tag{1}$$

## 3.2 Overestimation Bias in Deep Deterministic Policy Gradient

The Deep Deterministic Policy Gradient (DDPG) algorithm [45; 29] forms the foundation of various continuous control tasks such as TD3 [14] and SAC [22]. By enabling policy output to produce deterministic actions, DDPG offers a powerful algorithmic solution to these tasks. DDPG utilizes two neural networks: the actor and critic networks. The actor network is responsible for generating the policy, while the critic network evaluates the value function. In deep reinforcement learning (DRL), the policy $\pi(a|s)$ and the value function $Q(s, a)$ are expressed by deep neural networks parameterized with $\phi$ and $\theta$ respectively. DDPG suggests updating the policy through a deterministic policy gradient, as follows:

$$\nabla_\phi J(\phi) = \mathbb{E}_{s \sim P_\pi} \left[ \nabla_a Q^\pi(s, a)|_{a=\pi(s)} \nabla_\phi \pi_\phi(s) \right] \tag{2}$$

In order to approximate the discounted cumulative reward (as specified in Eq.(1)), the function $Q(s, a)$ is updated by minimizing the temporal difference (TD) errors [46] between the estimated value of the next state $s'$ and the current state $s$.

$$\theta^* = \arg \min_\theta \mathbb{E} \left[ r(s, a) + \gamma Q_\theta^\pi(s', a') - Q_\theta^\pi(s, a) \right]^2 \tag{3}$$

In policy evaluation, $Q$ function is used to approximate the value by taking action $a$ in state $s$. The $Q$ function is updated by Bellman equation [47] with bootstrapping, where we minimize the expected Temporal-Difference (TD) [46] error $\delta(s, a)$ between value and the target estimate in a mini-batch of samples $\mathcal{B}$,

$$\delta(s, a) = \mathbb{E}_\mathcal{B} \left[ r(s, a) + \max_{a'} Q(s', a') - Q(s, a) \right] \tag{4}$$

This process can lead to a consistent overestimation bias as the holding of Jensen's inequality [48]. Specifically, the overestimation of value evaluation arises due to the following process:

$$\mathbb{E}_\mathcal{B}[\max_{a' \in \mathcal{A}} Q(s', a')] \geq \max_{a' \in \mathcal{A}} \mathbb{E}_\mathcal{B}[Q(s', a')] \tag{5}$$

# 4 Methodology

This section describes **T**rajectories-awar**E** **E**nsemble exploratio**N** (TEEN) method that achieves diverse exploration for agents. TEEN aims to enforce efficient exploration by encouraging diverse behaviors of an ensemble of $N$ sub-policies parameterized by $\{\phi_i\}_{i=1}^N$. TEEN initially formulates a discrepancy measure and enforces the discrepancy among the sub-policies while simultaneously maximizing the expected return. Next, we present how to solve this optimization problem indirectly by utilizing the mutual information theory. Finally, we conduct a theoretical analysis to ensure whether the diversity of the samples collected by the ensemble policy is enhanced by solving this optimization problem. We summarize TENN in Algorithm 1.

## 4.1 Discrepancy Measure

We assess the diverse behaviors of policies by examining the diversity of trajectories generated by their interactions with the environments. The state-action visit distribution, indicating the probability of encountering a specific state-action pair when initiating the policy from a starting state, encapsulates the diversity of these trajectories directly. Thus, TEEN measures the diverse behaviors of policies by directly assessing their state-action visit distribution. Formally, given an ensemble of $N$ sub-policies $\{\pi_0, \pi_1, ..., \pi_k, ..., \pi_N\}$, we use $\rho^{\pi_k}$ to induce the state-action visit distribution deduced by policy $\pi_k$ and use $\rho$ for the ensemble policy. We have,

$$\rho = \frac{1}{N} \sum_{k=1}^{N} \rho^{\pi_k} \tag{6}$$

For convenience, we use the conditioned state-action visit probability, denoted as $\rho(s, a|z)$, to represent the respective state-action visit distribution of each sub-policy. This can be expressed as follows,

$$\rho^{\pi_k}(s, a) = \rho(s, a|z_k) \tag{7}$$

Where $\{z_k\}^N$ is the latent variable representing each policy. Our discrepancy measure $\mathcal{KL}$-divergence is an asymmetric measure of the difference between two probability distributions. Given this discrepancy measure, we define the difference of policies as the $\mathcal{KL}$-divergence of the conditioned state-action visit distribution with the state-action visit distribution.

**Definition 1** (Difference between Policies) Given an ensemble of $N$ policies $\{\pi_0, \pi_1, ..., \pi_k, ..., \pi_N\}$, the difference between policy $\pi_k$ and other policies can be defined by the conditioned state-action occupation discrepancy:

$$\mathcal{D}_{\mathcal{KL}}[\rho^{\pi_k}||\rho] := \mathcal{D}_{\mathcal{KL}}[\rho(s, a|z_k)||\rho(s, a)] \tag{8}$$

Consequently, we improve the difference between the conditioned state-action visit distribution to enforce diverse behaviors of sub-polices. While directly optimizing the Eq. (8) can be hard: Obtaining the state-action visit distribution can be quite challenging. In response to this, Liu, H., et al., as referenced in citation [49], have suggested the use of K-Nearest Neighbors (KNN) as an approximate measure for this distribution. We notice that it can be expressed by the form of entropy discrepancy:

$$\mathbb{E}_z\left[\mathcal{D}_{\mathcal{KL}}\left(\rho(s, a|z)||\rho(s, a)\right)\right] = \mathcal{H}(\rho) - \mathcal{H}(\rho|z) \tag{9}$$

In which the $\mathcal{H}[\cdot]$ is the Shannon Entropy with base $e$, $\mathcal{H}(\rho)$ is the entropy of the state-action visit distribution. With the mutual information theory, we can transform this optimization target as follows:

$$\mathcal{H}(\rho) - \mathcal{H}(\rho|z) = \mathcal{H}(z) - \mathcal{H}(z|\rho) \tag{10}$$

Thus, we turn to this equivalent optimization target. The first term encourages a high entropy of $p(z)$. We fix $p(z)$ to be uniform in this approach by randomly selecting one of the sub-policies to explore. We have $\mathcal{H}(z) = -\frac{1}{N}\Sigma_{k=1}^{N} \log p(z_k) \approx \log N$, which is a constant. The second term emphasises the sub-policy is easy to infer given a state-action pair. We approximate this posterior with a learned discriminator $q_\zeta(z|s, a)$ and optimize the variation lower bound.

$$\begin{aligned}
\mathcal{H}(z) - \mathcal{H}(z|\rho) &= \log N + \mathbb{E}_{s,a,z}[\log \rho(z|s, a)] \\
&= \log N + \mathbb{E}_{s,a}\left[\mathcal{D}_{\mathcal{KL}}(\rho(z|s, a)||q_\zeta(z|s, a))\right] + \mathbb{E}_{s,a,z}[\log q_\zeta(z|s, a))] \\
&\geq \log N + \mathbb{E}_{s,a,z}[\log q_\zeta(z|s, a))]
\end{aligned} \tag{11}$$

Where we use non-negativity of KL-divergence, that is $\mathcal{D}_{\mathcal{KL}} \geq 0$. We minimize $\mathcal{D}_{\mathcal{KL}}[\rho(z|s, a)||q_\zeta(z|s, a)]$ with respect to parameter $\zeta$ to tighten the variation lower bound:

$$\nabla_\zeta \mathbb{E}_{s,a}[\mathcal{D}_{\mathcal{KL}}(\rho(z|s, a)||q_\zeta(z|s, a))] = -\mathbb{E}_{s,a}[\nabla_\zeta \log q_\zeta(z|s, a)] \tag{12}$$

Consequently, we have the equivalent policy gradient with regularizer:

$$\pi^* = \arg \max_{\pi \in \Pi} J(\pi) + \alpha \mathbb{E}_{(s,a,z)\sim\rho}[\log q_\zeta(z|s, a))] \tag{13}$$

## 4.2 Trajectories-aware Ensemble Exploration

We select TD3 [14] as the algorithm of sub-policies which is not a powerful exploration algorithm. We then show how to ensemble algorithm with poor exploration performance and encourage efficient ensemble exploration by solving this optimization problem. We consider an ensemble of $N$ TD3 agents i.e., $\{Q_{\theta_k}, \pi_{\phi_k}\}_{k=1}^N$, where $\theta_k$ and $\phi_k$ denote the parameters of $k$-th Q-function and sub-policy. Combined with deterministic policy gradient method [29], we update the each sub-policy by policy gradient with regularizer.

$$\nabla J_{total}(\phi_k) = \mathbb{E}_{s \sim \rho}[\nabla_a(Q^\pi(s,a) + \alpha \log q_\zeta(z_k|s,a))|_{a=\pi_{\phi_k}(s)} \nabla_{\phi_k} \pi_{\phi_k}(s)] \quad (14)$$

**Recurrent Optimization.** Updating all sub-policies every time step in parallel may suffer from the exploration degradation problem [35]. Consider a state-action pair $(s, a_1)$, which is frequently visited by sub-policy $\pi_k$ and another state-action pair $(s, a_2)$ which is visited by multiple sub-policies such that we have $q_\zeta(z_k|s, a_1) > q_\zeta(z_k|s, a_2)$. Under this circumstance, the constraint encourage sub-policy $\pi_k$ to execute action $a_1$ while preventing from executing action $a_2$ which leads to the problem that some explored state-actions are prevented from being visited by the sub-policies. To tackle this challenge, we employ a recurrent training method for our sub-policies. In particular, given an ensemble of $N$ sub-policies, we randomly select a single sub-policy, denoted as $\pi_k$, every $T$ episodes. We then concentrate our efforts on regulating the behavior of the selected sub-policy, rather than all sub-policies simultaneously..

**Gradient Clip.** TEEN uses $\alpha \mathbb{E}_{(s,a,z) \sim \rho}[\log q_\zeta(z|s,a))]$ as a constraint to optimize the policy, while the gradient of this term is extremely large when the probability $q_\zeta(z|s,a)$ is small. And a state-action with small probability to infer the corresponding sub-policy $z_k$ implies that the state-action is rarely visited by sub-policy $z_k$ but frequently visited by other sub-policies. The target of the constraint is to increase the discrepancy of the sub-policies, while making the sub-policy visit this state-action will instead reduce the discrepancy. Further, for a state-action with large probability $q_\zeta(z|s,a)$, continuing to increase this probability will prevent the sub-policy from exploring other possible state-actions. Thus, we use the clipped constraint and the main objective we propose is the following:

$$\pi^* = \arg \max_{\pi \in \Pi} J(\pi) + \alpha \mathbb{E}_{s,a,z}[\log \text{clip}(q_\zeta(z|s,a), \epsilon, 1 - \epsilon))] \quad (15)$$

where $\epsilon$ is a hyper-parameter, and we set $\epsilon = 0.1$ for all of our experiments.

## 4.3 Controlling the Estimation Bias

In Q-value estimation, we share some similarities with Max-Min DQN [11] and Averaged DQN [50]. While Max-Min DQN takes the minimum over an ensemble of value functions as the target encouraging underestimation, Averaged DQN reduces the variance of estimates by using the mean of the previously learned Q-values. We combine these two techniques and apply both mean and min operators to reduce overestimation bias and variance. Instead of using the mean of the previously learned Q-values, we use the mean value on $N$ actions of $N$ sub-policies. TEEN then uses the minimum over a random subset $\mathcal{M}$ of $M$ Q-functions as the target estimate.

$$Q^{target} = r(s,a) + \gamma \min_{i=1,2,...,M} \frac{1}{N} \sum_{j=1}^N Q_i(s, \pi_j(s)) \quad (16)$$

We give some analytical evidences from the perspective of order statistics to show how we control the estimation bias by adjusting $M$ and $N$.

**Theorem 1.** Let $X_1, X_2, ..., X_N$ be an infinite sequence of i.i.d. random variables with a probability density function (PDF) of $f(x)$ and a cumulative distribution function (CDF) of $F(x)$. Denote $\mu = \mathbb{E}[X_i]$ and $\sigma^2 = \mathbb{V}[X_i]$. Let $X_{1:N} \leq X_{2:N} \leq X_{3:N}... \leq X_{N:N}$ be the order statistics corresponding to $\{X_i\}_N$. Denote PDF and CDF of the $k$-th order statistic $X_{k:N}$ as $f_{k:N}$ and $F_{k:N}$ respectively. The following statements hold.

(i) $\mu - \frac{(N-1)\sigma}{\sqrt{2N-1}} \leq \mathbb{E}[X_{1:N}] \leq \mu, N > 1$. $\mathbb{E}[X_{1:N+1}] \leq \mathbb{E}[X_{1:N}]$

---

**Algorithm 1** Trajectories-Aware Ensemble Exploration algorithm (TEEN)

---

1: Initialize $N$ policy parameters $\phi_k$, $N$ Q-function parameters $\theta_k$, $k = 1, 2, ..., N$
2: Initialize target Q-function parameters $\theta'_k \leftarrow \theta_k, k = 1, 2, ..., N$.
3: Initialize discriminator $q_\zeta$
4: Initialize empty replay buffer $\mathcal{D}$
5: Initialize recurrent interval $d$
6: **for** each iterations **do**
7:      Randomly sample a policy $\pi_k$
8:      **for** each time step $t$ **do**
9:          Choose one action with noise $a_t = \pi_{\phi_k}(s) + \epsilon, \epsilon \sim \mathcal{N}(0, \sigma)$
10:          Collect state $s_{t+1}$ and reward $r_t$ from the environment by taking action $a_t$
11:          Store transition $\tau_t = (s_t, a_t, s_{t+1}, r_t)$ in replay buffer $\mathcal{B} \leftarrow \mathcal{B} \cup \tau_t$
12:      **end for**
13:      **if** $t$ mod $d$ **then**
14:          Randomly select one sub-policy $\pi_{\phi_i}$
15:      **end if**
16:      **for** each gradient step **do**
17:          Sample random minibatch $\mathcal{B} = \{\tau_t\}^B \sim \mathcal{D}$
18:          Randomly select a set $\mathcal{M}$ of M distinct indices from $\{1, 2, ...N\}$.
19:          Compute the $Q$ target $y$ by Eq. (16)
20:          For all of the value functions, update $\theta_k$ using gradient decent
21:
$$\nabla_{\theta_k} \frac{1}{|B|} \sum_{(s,a,r,s') \in \mathcal{B}} (y - Q_{\theta_k}(s, a))^2$$
22:          For sub-policy $\pi_{\phi_i}$, update $\phi_i$ by gradient decent
23:
$$\nabla_{\phi_i} \frac{1}{|B|} \sum_{(s,a,r,s') \in \mathcal{B}} (\alpha \log q_\zeta(z_i|s, a) - Q_{\theta_i}(s, a)), \quad a = \pi_{\phi_i}(s)$$
24:          Update target Q-function by $\theta'_k \leftarrow \rho\theta_k + (1 - \rho)\theta'_k$
25:          Update discriminator $q_\zeta$ by Eq. (12)
26:      **end for**
27: **end for**

---

(ii) Let $\bar{X} = \frac{1}{N} \Sigma_{i=1}^N X_i$, then, $\mathbb{E}[\bar{X}] = \mu, Var[\bar{X}] = \frac{1}{N}\sigma^2$

Points 1 of the Theorem indicates that the minimum over an ensemble of values reduces the expected value, which implies that we can control the estimation from over estimates to under or proper estimates. The second point indicates that the mean over an ensemble of values reduces the variance. Thus, by combining these two operators, we control the estimation bias and the variance.

### 4.4 Theoretical Analysis

We analyze the diversity of state-actions gathered by ensemble policy on the basis of Shannon entropy $\mathcal{H}[\cdot]$, a commonly used diversity measure [32; 34; 51], which is a measure of the dispersion of a distribution. A high entropy of the state-action visit distribution implies that the trajectory distribution is more dispersed, which means more diverse trajectories. Thus, to achieve diverse trajectories, we maximize the expected return while maximizing the entropy of the state-action visit distribution:

$$\pi^* = \arg\max_{\pi \in \Pi} J(\pi) + \alpha\mathcal{H}[\rho^\pi] \tag{17}$$

Where $\rho^\pi$ deduce the state-action visit distribution induced by the ensemble policy $\pi$ and $\alpha$ is the weighting factor. The diversity of trajectories gathered by the ensemble policy comes from two

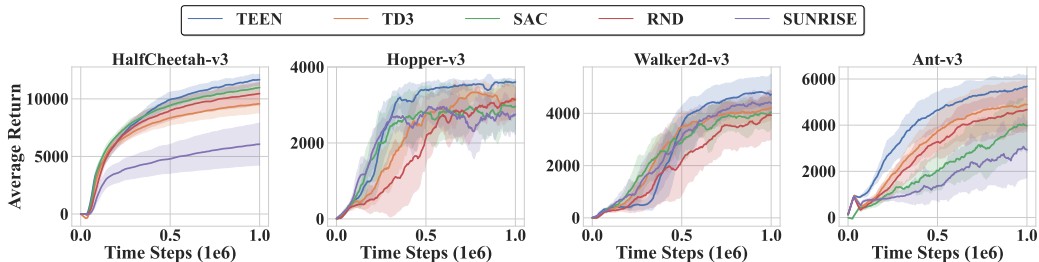

Figure 1: Learning curves for 4 MuJoCo continuous control tasks. For better visualization, the curves are smoothed uniformly. The bolded line represents the average evaluation over 10 seeds. The shaded region represents a standard deviation of the average evaluation over 10 seeds.

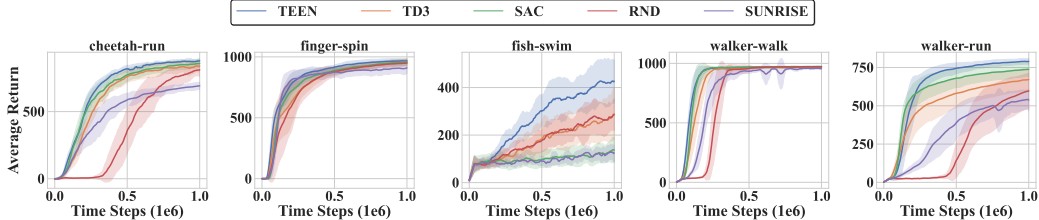

Figure 2: Learning curves for 5 continuous control tasks on DMControl suite. For better visualization, the curves are smoothed uniformly. The bolded line represents the average evaluation over 10 seeds. The shaded region represents the standard deviation of the average evaluation over 10 seeds.

components: the discrepancy of sub-policies and the diversity of trajectories gathered by the sub-policies. To illustrate that,

**Lemma 1** (Ensemble Sample Diversity Decomposition) Given the state-action visit distribution of the ensemble policy $\rho$. The entropy of this distribution is $\mathcal{H}(\rho)$. Notice that this term can be decomposed into two parts:

$$\mathcal{H}(\rho) = \mathbb{E}_z[\mathcal{D}_{\mathcal{KL}}(\rho(s,a|z_k)||\rho(s,a))] + \mathcal{H}(\rho|z) \tag{18}$$

Where the first term is the state-action visit distribution discrepancy between the sub-policies and the ensemble policy induced by $\mathcal{KL}$-divergence measure. The second term implies the diversity of state-action visited by sub-policies which depends on which algorithm is used for the sub-policy, such as TD3 [14], SAC [22], RND [24], etc.

As shown in this inequality, $\mathcal{H}(\rho)$ is irrelevant of ensemble size $N$. Therefore, the ensemble size may improve the diversity of the ensemble policy by influencing the discrepancy of sub-policies indirectly, which implies that the diversity is not guaranteed with the increased ensemble size. Our method increases the discrepancy of sub-policies which theoretically improve the sample diversity of the ensemble policy.

## 5 Experimental Setup

This section presents the experimental setup used to evaluate our proposed TEEN and assesses its performance in the MuJoCo environments.

**Environments.** We evaluate our algorithm on several continuous control tasks from MuJoCo control suite [25] and DeepMind Control Suite [26]. We conduct experiments on 4 control tasks in MuJoCo control suite namely HalfCheetah-v3, Hopper-v3, Walker2d-v3, Ant-v3. In Halfcheetah-v3, the task is to control a cheetah with only one forefoot and one hind foot to run forward. In Hopper-v3, we control a single-legged robot to hop forward and keep balance when hopping. Walker2d-v3 is a bipedal robot environment, where we train the agent to walk or run forward. The target of Ant-v3 is to train a quadrupedal agent to stay balance and move forward. These four environments are all challenging continuous control tasks in MuJoCo. On DeepMind Control Suite, we evaluate our algorithm in 5 environments: cheetah-run, finger-spin, fish-swim, walker-walk, walker-run.

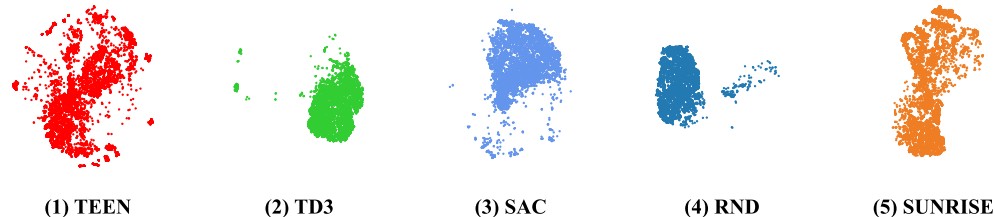

(1) TEEN      (2) TD3      (3) SAC      (4) RND      (5) SUNRISE

Figure 3: Measuring the exploration region. The points represent region explored by each method in 10 episodes. All the states get dimension reduction by the same t-SNE transformation for better visualization.

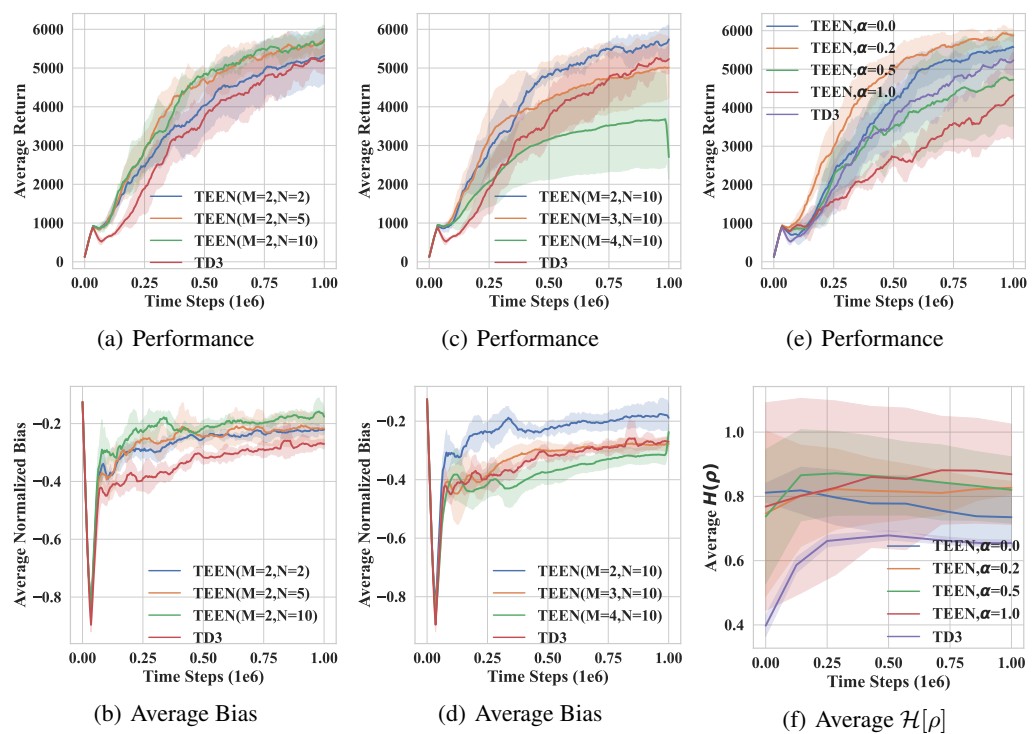

Figure 4: TEEN ablation results for Ant environment. The first column shows the effect of ensemble size $N$ on both performance and estimation bias. The second column shows the effect of target value number $M$ on both performance and estimation bias. The third column shows the effect of weight parameter $\alpha$ on both performance and the entropy of trajectories.

**Baselines.** Beyond simply comparing our approach with the foundational algorithm TD3 [14], a leading-edge deterministic policy gradient reinforcement learning algorithm, we also contrast our method with other efficient exploration algorithms across three distinct categories. We examine curiosity-based exploration baselines, such as RND [24], which promotes efficient exploration through intrinsic curiosity rewards. Regarding maximum entropy-based exploration, we consider SAC [22], an off-policy deep reinforcement learning algorithm that offers sample-efficient exploration by maximizing entropy during training. For ensemble exploration, we look at SUNRISE [16], a unified framework designed for ensemble-based deep reinforcement learning.

**Evaluation.** We follow the standard evaluation settings, carrying out experiments over one million (1e6) steps and running all baselines with ten random seeds per task to produce the main results. We maintain consistent learning rates and update ratios across all baselines. Further details regarding these parameters can be found in the Appendx B.2. For replication of TD3, SAC and SUNRISE, we use the code the author released for each baseline without any modifications to ensure the performance. For the reproduction of RND, we follow to the repository provided in the original code. For a fair

Table 1: Performance on MuJoCo Control Suite at 500K and 1M timesteps. The results show the average returns with mean and variance over 10 runs. The maximum value for each task is bolded.

| 500K Step | HalfCheetah | Hopper | Walker2d | Ant |
|---|---|---|---|---|
| TEEN | **10139±623** | **3548±100** | **4034±547** | **5009±649** |
| TD3 | 8508±601 | 3330±135 | 3798±511 | 4179±809 |
| SAC | 9590±419 | 3332±223 | 3781±521 | 3302±798 |
| RND | 9185±813 | 2814±536 | 2709±1341 | 3666±685 |
| SUNRISE | 4955±1249 | 3426±99 | 3782±516 | 1964±1020 |
| 1M Step | | | | |
| TEEN | **11914 ± 448** | **3687 ± 57** | **5099±513** | **5930 ± 486** |
| TD3 | 9759±778 | 3479±147 | 4229±468 | 5142 ± 940 |
| SAC | 11129±420 | 3484±128 | 4349±567 | 5084±901 |
| RND | 10629±942 | 3148±143 | 4197±791 | 4990±789 |
| SUNRISE | 6269±1809 | 3644±75 | 4819±398 | 3523±1430 |

comparison, we use fully connected network with 3 layers as the architecture of the critic networks and actor networks. The learning rate for each network is set to be $3e-4$ with Adam [52] as the optimizer. The implementation details for each baseline can be found in Appendx B.2.

## 6   Experimental Results

We conducted experiments to evaluate our algorithm's performance in addressing three key research questions (RQ). Specifically, we sought to determine whether TEEN outperforms the current state-of-the-art across multiple environments and achieves a boarder range of exploration region, as well as how the ensemble size and discrepancy of sub-policies (defined in Eq. 9) impact TEEN's performance.

**RQ1: Whether TEEN outperforms the current state-of-the-art across multiple environments?**
Table 1 and Table 2 report the average returns with mean and variance of evaluation roll-outs across all algorithms. The learning curves during training on MuJoCo control suite are shown in Figure 1. Learning curves for DeepMind Control suite can be found in Figure 2. On MuJoCo continuous control suite, TEEN shows superior performance across all environments which implies that by ensemble sub-policies with weak exploration performance, such as TD3, the exploration performance of the ensemble policy enhances evenly exceeding algorithms with strong exploration performance, SAC, RND, etc. While purely ensembling multiple models may not certainly improve the performance. Although SUNRISE improves the exploration performance by UCB exploration [53], it does not take effect when the uncertainty shows little relevance to the environment, and even degrades the performance (as shown in HalfCheetah-v3). TEEN can be seen as an immediate solution for diverse exploration within ensemble reinforcement learning by directly enforcing an ensemble policy to discover a broader range of trajectories. We further conduct experiments to validate that TEEN achieves a boarder exploration region (Figure 3). For a fair comparison, all the algorithms are trained in Ant-v3 with the same seed at half of the training process. In order to get reliable results, the states explored are gathered in 10 episodes with different seeds. We apply the same t-SNE transformation to the states explored by all of the algorithms for better visualization.

**RQ2: How the ensemble size $N$ and number of target values $M$ impact TEEN's performance?**
We make experiments in Ant-v3 environment from OpenAI gym. As shown in Figure 3, when using $M = 2$ as the number of target values, the estimate of the target value is underestimated and the estimation bias increases with the increase of $M$. This underestimation is mitigated with an increase in ensemble size $N$ resulting in better performance. Thus, we choose 10 to be the ensemble size for all the environments. To enhance reliability, we conducted ablation studies in various MuJoCo

Table 2: Performance on DeepMind Control Suite at 500K and 1M timesteps. The results show the average returns with mean and variance over 10 runs. The maximum value for each task is bolded.

| 500K Step | Cheetah-Run | Finger-Spin | Fish-Swim | Walker-Walk | Walker-Run |
|---|---|---|---|---|---|
| TEEN | **837±23** | **928±22** | **381±71** | 970±6 | **754±33** |
| TD3 | 784±50 | 891±24 | 260±77 | 968±5 | 592±87 |
| SAC | 791±46 | 897±30 | 155±34 | **972±4** | 688±55 |
| RND | 408±159 | 894±32 | 254±42 | 963±20 | 180±107 |
| SUNRISE | 606±37 | 885±38 | 201±34 | 954±16 | 406±99 |
| 1M Step | | | | | |
| TEEN | **891±7** | **976±12** | **519±79** | 974±5 | **800±16** |
| TD3 | 863±29 | 965±22 | 367±113 | 972±5 | 683±69 |
| SAC | 871±23 | 971±22 | 202±50 | **976±3** | 746±42 |
| RND | 841±35 | 960±24 | 385±66 | 975±3 | 606±110 |
| SUNRISE | 703±26 | 920±40 | 285±46 | 969±4 | 553±54 |

environments, obtaining consistent results. The results for other environments can be found in the Appendix C.

**RQ3: How the discrepancy of sub-policies impact TEEN's performance?** In order to validate the benefits of trajectory-aware exploration, we conduct experiments in the Ant-v3 environment, progressively increasing the weight parameter denoted as $\alpha$. We adjust the value of the weight $\alpha$ from $[0, 0.2, 0.5, 1]$. With $\alpha = 0$, we show the performance without trajectory-aware exploration. As the results presented in Figure 3, with adequate reward scale, TEEN improves the performance with the use of trajectory-aware exploration. The effectiveness of the approach fluctuates with changes in reward scale. For a large $\alpha$ value, TEEN aims to enhance the diversity of trajectories. However, as $\alpha$ increases, TEEN fails to effectively utilize reward signals, resulting in sub-optimal performance. In the experiment, we used $\alpha = 0.2$ for the MuJoCo control suites. However, the reward scales in the DeepMind control suites are significantly smaller than those in the MuJoCo control suites. In the MuJoCo control suite, the accumulated reward exceeds 5,000 in the Ant environment, while in the DeepMind control suite, the maximum accumulated reward for all tasks is 1,000. As a result, we utilized a smaller $\alpha = 0.02$ for the DeepMind control suite, considering the smaller reward scale. We also conducted ablation studies with varying reward scales across different environments, please refer to Appendix C for more details.

## 7 Conclusion

This paper presents a theoretical analysis of the ensemble policy's diversity exploration problem, based on trajectory distribution. We innovatively discover that the diversity of the ensemble policy can only be enhanced by increasing the Kullback-Leibler (KL) divergence of the sub-policies when entropy is used as the measure of diversity. Guided by this analysis, we introduce the Trajectories-aware Ensemble Exploration (TEEN) method — a novel approach that encourages diverse exploration by improving the diversity of trajectories. Our experiments demonstrate that TEEN consistently elevates the performance of existing off-policy RL algorithms such as TD3, SAC, RND, and ensemble RL algorithms like SUNRISE. There are still important issues to be resolved, e.g., TEEN is sensitive to the reward scale resulting in poor performance without a proper parameter setting. We hope these will be addressed by the community in the future.

## Acknowledgments

This work was funded by the National Key R&D Program of China (2021YFC2800500).

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
