## A   Theoretical Derivations

**Lemma 1** (Ensemble Sample Diversity Decomposition) Given the state-action visit distribution of the ensemble policy $\rho$. The entropy of this distribution is $\mathcal{H}(\rho)$. Notice that this term can be decomposed into two parts:

$$\mathcal{H}(\rho) = \mathbb{E}_z[\mathcal{D}_{\mathcal{KL}}(\rho(s,a|z_k)||\rho(s,a))] + \mathcal{H}(\rho|z) \tag{1}$$

*Proof.*

$$\begin{aligned}
\mathcal{H}(\rho) &= \mathbb{E}_{(s,a)\sim\rho}[-\log(\rho(s,a))] \\
&= \mathbb{E}_{(s,a,z)\sim\rho}\left[\log\frac{\rho(s,a|z)}{\rho(s,a)} - \log\rho(s,a|z)\right] \\
&= \mathbb{E}_z\left[\mathcal{D}_{\mathcal{KL}}(\rho(s,a|z)||\rho(s,a))\right] + \mathcal{H}(\rho|z)
\end{aligned} \tag{2}$$

$\square$

**Lemma 2**(Equivalent optimization target)

$$\mathcal{H}(z|\rho) \propto -\mathcal{D}_{\mathcal{KL}}\left[\rho(s,a|z)||\frac{1}{n}\sum_{k=1}^{n}\rho(s,a|z_k)\right] \tag{3}$$

*Proof.* By definition,

$$\mathcal{I}(\rho;z) = \mathcal{H}(\rho) - \mathcal{H}(\rho|z) = \mathcal{H}(z) - \mathcal{H}(z|\rho) \tag{4}$$

By randomly selecting the latent variable $z$, we consider that $\mathcal{H}(z)$ is a constant depending on the number of $z$. Thus, we have,

$$\begin{aligned}
\mathcal{H}(z|\rho) &= \mathcal{H}(z) + \mathcal{H}(\rho|z) - \mathcal{H}(\rho) \\
&\propto \mathbb{E}_{(s,a,z)\sim\rho(s,a,z)}[-\log\rho(s,a|z)] - \mathbb{E}_{(s,a)\sim\rho(s,a)}[-\log\rho(s,a)] \\
&= \mathbb{E}_{(s,a,z)\sim\rho(s,a,z)}[-\log\rho(s,a|z)] - \int -\rho(s,a)\log\rho(s,a)dsda \\
&= \mathbb{E}_{(s,a,z)\sim\rho(s,a,z)}[-\log\rho(s,a|z)] - \int -\rho(s,a,z)\log\rho(s,a)dsdadz \\
&= \mathbb{E}_{(s,a,z)\sim\rho(s,a,z)}[\log\rho(s,a) - \log\rho(s,a|z)]
\end{aligned} \tag{5}$$

Where, $\rho(s,a) = \int \rho(s,a|z)p(z)dz = \frac{1}{n}\sum_{k=1}^{n}\rho(s,a|z_k)$.
Then,

$$\begin{aligned}
\mathcal{H}(z|\rho) &= \mathbb{E}_{(s,a,z)\sim\rho(s,a,z)}[-\log\rho(z|s,a)] \\
&= \mathcal{H}(z) + \mathcal{H}(\rho|z) - \mathcal{H}(\rho) \\
&\propto \mathbb{E}_{(s,a,z)\sim\rho(s,a,z)}\left[\frac{1}{n}\sum_{k=1}^{n}\rho(s,a|z_k) - \log\rho(s,a|z)\right] \\
&\propto -\mathcal{D}_{\mathcal{KL}}\left[\rho(s,a|z)||\frac{1}{n}\sum_{k=1}^{n}\rho(s,a|z_k)\right]
\end{aligned} \tag{6}$$

$\square$

**Lemma 3** Let $X_1, X_2, ..., X_N$ be an infinite sequence of i.i.d. random variables with a probability density function (PDF) of $f(x)$ and a cumulative distribution function (CDF) of $F(x)$. Let $X_{1:N} \leq X_{2:N} \leq X_{3:N}... \leq X_{N:N}$ be the order statistics corresponding to $\{X_i\}_N$. Denote PDF and CDF of the $k$-th order statistic $X_{k:N}$ as $f_{k:N}$ and $F_{k:N}$ respectively. The following statements hold.

(i) $F_{N:N}(x) = (F(x))^N$. $f_{N:N}(x) = Nf(x)(F(x))^{N-1}$

(ii) $F_{1:N}(x) = 1 - (1-F(x))^N$. $f_{1:N}(x) = Nf(x)(1-F(x))^{N-1}$

(iii) $\mu - \frac{(N-1)\sigma}{\sqrt{2N-1}} \le \mathbb{E}[X_{1:N}] \le \mu, N > 1.$ $\mathbb{E}[X_{1:N+1}] \le \mathbb{E}[X_{1:N}]$

(iv) Let $\bar{X} = \frac{1}{N}\Sigma_{i=1}^{N}X_i$, then, $\mathbb{E}[\bar{X}] = \mu, Var[\bar{X}] = \frac{1}{N}\sigma^2$

*Proof.* (i) We start from the CDF of $X_{N:N}$. By definition, $F_{N:N}(x) = P(X_{N:N} \le x) = P(X_1 \le x, X_2 \le x, ..., X_N \le x)$. Under the assumption of iid. $P(X_1 \le x, X_2 \le x, ..., X_N \le x) = P(X_1 \le x)P(X_2 \le x)...P(X_N \le x) = (F(x))^N$. The PDF of $X_{N:N}$ can be derived by taking the derivative of PDF. $f_{N:N} = \frac{dF_{N:N}(x)}{dx} = Nf(x)(F(x))^{N-1}$.

(ii) Similar to (i), $F_{1:N}(x) = P(X_{1:N} \le x) = 1 - P(x \le X_{1:N}) = P(x \le X)P(x \le X_1, x \le X_2, ..., x \le X_N)$. Under the assumption of iid. $P(x \le X_1, x \le X_2, ..., x \le X_N) = P(X_1 \ge x)P(X_2 \ge x)...P(X_N \ge x)$. Satisfying the normalization, we have $P(X_1 \ge x)P(X_2 \ge x)...P(X_N \ge x) = (1 - P(X_1 \le x))(1 - P(X_2 \le x))...(1 - P(X_N \le x)) = (1 - F(x))^N$. Thus, $F_{1:N}(x) = 1 - (1 - F(x))^N$. By taking the derivative of PDF, $f_{1:N}(x) = Nf(x)(1 - F(x))^{N-1}$.

(iii) The detailed proof can be found in [54]. An brief proof is provided as follows. By definition,

$$
\begin{aligned}
\mathbb{E}[X_{1:N}] &= \int_{-\infty}^{+\infty} xf_{1:N}(x)dx \\
&\overset{(i)}{=} \int_{-\infty}^{+\infty} xNf(x)(1 - F(x))^{N-1}dx \\
&\overset{u=F(x)}{=} \int_0^1 x(u)N(1 - u)^{N-1}du
\end{aligned}
\tag{7}
$$

To obtain the lower bound on $\mathbb{E}[X_{1:N}]$, we consider the extremum of $\mathbb{E}[X_{1:N}]$ and constrains of mean and variance. For simplification, we consider zero-mean distribution with $\mu = 0$ and $\sigma^2$. The lower bound can be obtained by applying Cauchy-Buniakowsky-Schwarz inequality,

$$
\left(\int_0^1 x(u)N(1 - u)^{N-1}du\right)^2 \le \int_0^1 x^2 du \int_0^1 (N(1 - u)^{N-1})^2 du = \frac{(N-1)^2}{2N-1}\sigma^2, N > 1 \tag{8}
$$

Thus, we have $\mathbb{E}[X_{1:N}] \ge \mu - \frac{N-1}{\sqrt{2N-1}}\sigma$ for distribution with mean and variance of $\mu$ and $\sigma$ respectively. By definition, $\mathbb{E}[X_{1:N+1}] = \mathbb{E}[\min(X_{1:N}, X_{N+1})] \le \mathbb{E}[X_{1:N}]$

(iv) By definition,

$$
\mathbb{E}[\bar{X}] = \mathbb{E}\left[\frac{1}{N}\Sigma_{i=1}^N X_i\right] = \frac{1}{N}\Sigma_{i=1}^N \mathbb{E}[X_i] = \mu \tag{9}
$$

$$
\begin{aligned}
Var[\bar{X}] &= Var\left[\frac{1}{N}\Sigma_{i=1}^N X_i\right] \\
&= \mathbb{E}\left[\left(\frac{1}{N}\Sigma_{i=1}^N X_i\right)^2\right] - \mathbb{E}^2\left[\frac{1}{N}\Sigma_{i=1}^N X_i\right] \\
&= \frac{1}{N^2}\mathbb{E}\left[\Sigma_{i=1}^N \Sigma_{j=1}^N X_i X_j\right] - \mu^2 \\
&= \frac{1}{N^2}\mathbb{E}\left[\Sigma_{i=1}^N X_i^2\right] + \frac{1}{N^2}\Sigma_{i=1}^N \Sigma_{j=1, j\neq i}^N \mathbb{E}[X_i]\mathbb{E}[X_j] - \mu^2 \\
&= \frac{1}{N}(\mu^2 + \sigma^2) - \frac{1}{N}\mu^2 \\
&= \frac{1}{N}\sigma^2
\end{aligned}
\tag{10}
$$

$\square$

## B  Experimental Details

### B.1  Experimental Setup

To validate the generalization, we evaluate our proposed algorithm on environments from MuJoCo
Control Suite [25] and DMControl Suite [26]. We use the publicly available environments without
any modification. We present screenshots in Figure 1.

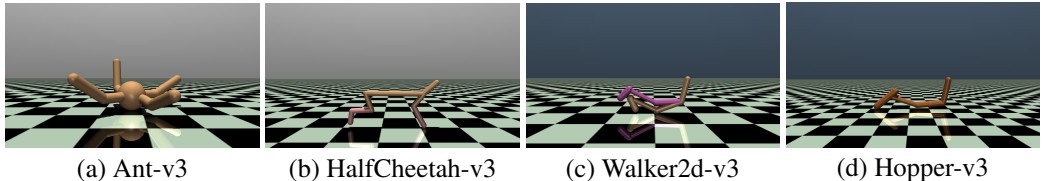

(a) Ant-v3          (b) HalfCheetah-v3          (c) Walker2d-v3          (d) Hopper-v3

Figure 1: Screenshots of MuJoCo environments.

### B.2  Implementation and Hyper-parameters

Here, we describe certain implementation details of TEEN. For our implementation of TEEN, we use
a combination of TD3 [14] and TEEN, where we construct $N$ TD3 agents based on the released code
by the autor (https://github.com/sfujim/TD3). We implement a total of $N = 10$ TD3 agents through
out our entire experiments. For recurrent optimization mentioned in section 4.2, we set the period of
recurrent training to be 50k. We provide explicit parameters used in our algorithm in Table 1.

### B.3  Reproducing Baselines

For reproduction of TD3, we use the official implementation ( https://github.com/sfujim/TD3). For
implementation of SAC, we use the code the author provided and use the parameters the author
recommended. We use a single Gaussian distribution and use the environment-dependent reward
scaling as described by the authors. For a fair comparison, we apply the version of soft target update
and train one iteration per time step.

Table 1: TEEN Parameters settings

| Parameter | Value |
|---|---|
| Exploration policy | $\mathcal{N}(0, 0.1)$ |
| Weight $\alpha$ | 0.2 |
| Number of sub-policies $N$ | 10 |
| Number of target values $M$ | 2 |
| Variance of exploration noise | 0.2 |
| Random starting exploration time steps | $2.5 \times 10^4$ |
| Optimizer | Adam[52] |
| Learning rate for actor | $3 \times 10^{-4}$ |
| Learning rate for critic | $3 \times 10^{-4}$ |
| Replay buffer size | $1 \times 10^6$ |
| Batch size | 256 |
| Discount ($\gamma$) | 0.99 |
| Number of hidden layers | 2 |
| Number of hidden units per layer | 256 |
| Activation function | ReLU |
| Iterations per time step | 1 |
| Target smoothing coefficient ($\eta$) | $5 \times 10^{-3}$ |
| Variance of target policy smoothing | 0.2 |
| Noise clip range | $[-0.5, 0.5]$ |
| Target critic update interval | 2 |

# C  Additional Experimental Results

## C.1  Additional Evaluation

To convince our evaluation, we conduct our algorithm in a challenging environment Humanoid-v3 in the MuJoCo suite, which is shown in Figure 2. The state dimension of Humaniod is 376, which is exceptionally difficult to solve. We compared our algorithm TEEN with sample-efficient algorithms TD3 and SAC. We follow the standard evaluation settings, carrying out experiments over five million (5e6) steps and running all baselines with 5 random seeds.

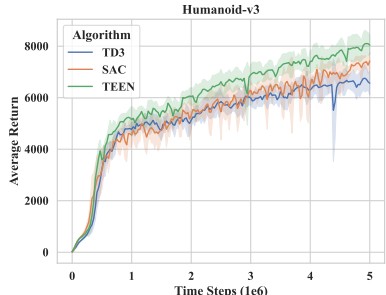

Figure 2: Learning curves for MuJoCo complex task Humanoid-v3. For better visualization,the curves are smoothed uniformly. The bolded line represents the average evaluation over 5 seeds. The shaded region represents a standard deviation of the average evaluation over 5 seeds.

## C.2  Additional Ablation Studies

We perform ablation studies on other environments in the MuJoCo suite and show the learning cuives in Figure 3. All the experiments are performed over 5 random seeds with one million (1e6) steps.

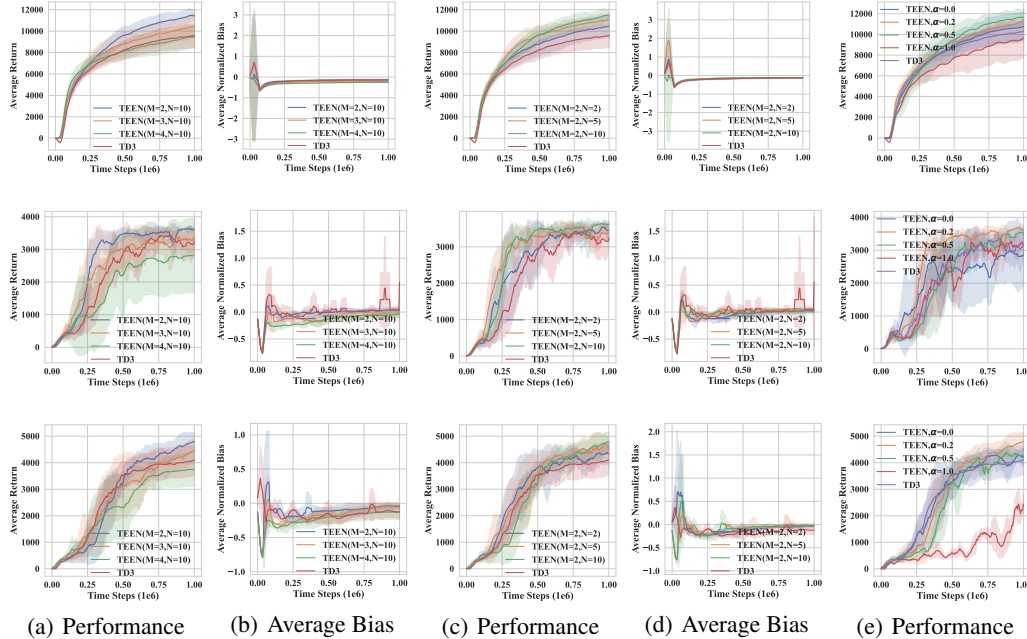

(a) Performance  (b) Average Bias  (c) Performance  (d) Average Bias  (e) Performance

Figure 3: TEEN ablation results for MuJoCo environments (From top to bottom are HalfCheetah-v3, Hopper-v3, Walker2d-v3.). The first column shows the effect of ensemble size $N$ on performance. The second column shows the effect of ensemble size $N$ on estimation bias. The third and fourth columns show the effect of target value number $M$ on both performance and estimation bias respectively. The fifth column shows the effect of weight parameter $\alpha$ on performance.