# OpenReview forum: "Keep Various Trajectories: Promoting Exploration of Ensemble Policies in Continuous Control"
_NeurIPS.cc/2023/Conference — NeurIPS 2023 poster_

### Official Review · Reviewer_vN15 · 2023-07-04

**Soundness:** 3 good
**Presentation:** 2 fair
**Contribution:** 2 fair
**Rating:** 6
**Confidence:** 3

**Summary:**

This paper addresses the problem of exploration in ensemble-based reinforcement learning. It introduces a novel approach called Trajectory Aware Ensemble Exploration (TEEN) that aims to enhance exploration by increasing diversity among the sub-policies. This diversity is measured by the KL divergence between the distributions of the trajectories. The method achieves diversity by optimizing the variational lower bound, which is then incorporated into the policy gradient step with the addition of a regularizer term.

To reduce estimation bias on the Q function, the method selects the minimum value from a random set of critics based on the average Q value of the actions chosen by all sub-policies. The paper provides theoretical analysis to support this design choice, demonstrating that it reduces both estimation bias and variance.

The effectiveness of TEEN is validated through experimental results on MuJoco and Deepmind Control Suite. These results show that the proposed method successfully explores a wide range of states and surpasses the performance of existing DRL algorithms in continuous control, including those specifically designed for efficient exploration.


**Strengths:**

- It proposes a new method that improves the exploration in ensemble-based RL by increasing the diversity of the sub-polices in terms of their state-action visit distribution.
- The implementation is straightforward and involves two steps on top of a DDPG-like method:
  - add a regularizer to the policy gradient
  - add a routine to select and update sub-policies and to compute the target Q value
- It provides theoretical analysis to justify the choice of target Q value
- Experimental results are presented to validate the efficacy of the method.


**Weaknesses:**

- There appears to be an inconsistency between Line 18 in Algorithm 1 and the text. Specifically, Line 18 suggests that all sub-policies are updated sequentially, while the "recurrent optimization" section indicates that only one sub-policy is updated in each gradient step. Could you clarify this discrepancy?

- Theorem 1 demonstrates that the method reduces the expected Q value. However, the experimental results indicate that the algorithm tends to underestimate the Q value. It's unclear whether underestimation is preferable to overestimation. If possible, it would be beneficial to address this aspect in the analysis.

- The main paper frequently refers to Appendix C, but it appears to be missing from the appendix section.

- In Line 436 of the appendix, it seems that the last expression is missing a multiplier p(s). Could you confirm?

- There's a mismatch between Lemmas 2 and 3 in the appendix and the content presented in the main paper.

- The proof for (v) in Lemma 3 does not show the variance inequality: V[X_{1:{N+1}}] \geq V[X_{1:{N}}].

- Formatting Issue:

  - The readability could be improved if Table 2 used the same row/column heading as those in Table 1.
  - Appendix line 483: Table index is missing.

- Grammar:
  - Line 151: we can “increase” this equivalent optimization target. “maximize” seems more appreciate here.
  - Line 272: “.While purely ensemble multiple models may not certainly improve the performance.“
  - Line 273: “it does not take effect“, “evenly degrading”.


**Questions:**

Please refer to the weakness section.

**Limitations:**

- The authors have discussed one limitation regarding the sensitivity about reward scaling in Sec. 7.
- Could the authors comment on the computational overhead of the ensemble?

---

> ### Author Rebuttal · Authors · 2023-08-05
>
> We thank you for your detailed feedback and insightful suggestions, please see the following for our response.
>
> > Q1: There appears to be an inconsistency between Line 18 in Algorithm 1 and the text. Specifically, Line 18 suggests that all sub-policies are updated sequentially, while the "recurrent optimization" section indicates that only one sub-policy is updated in each gradient step. Could you clarify this discrepancy?
>
> A1: Thanks for pointing this out. This line is incorrectly stated, we do update only one sub-policy in each gradient step with recurrent optimization. We will modify this part of the statement in the algorithm in the next version.
>
> > Q2: Theorem 1 demonstrates that the method reduces the expected Q value. However, the experimental results indicate that the algorithm tends to underestimate the Q value. It's unclear whether underestimation is preferable to overestimation. If possible, it would be beneficial to address this aspect in the analysis.
>
> A2: This is an open question. Underestimation may not always be preferable to overestimation for all tasks. However, in general, overestimation bias is usually considered more harmful than underestimation bias [1,2,3]. This is because usually overestimation bias directly destroys the learning of an intelligent body in severe cases [3], while underestimation bias can be mitigated by visiting the corresponding state-action pairs multiple times [1]. When dealing with underestimation bias, it is easy to introduce overestimation bias [1,2]. We follow the general way of dealing with overestimation bias. Our concern is how to make the value function estimate closer to the true estimate in this submission.
>
> > Q3: The main paper frequently refers to Appendix C, but it appears to be missing from the appendix section.
>
> A3: Sorry for that, and check the general rebuttal for the missing part.
>
> > Q4: In Line 436 of the appendix, it seems that the last expression is missing a multiplier $p(s)$. Could you confirm?
>
> A4: Sorry for this typo. We present the full proof of Lemma 2 here.
>
> Lemma 2
>
> Proof.
>
> By definition,
>
> $\mathcal{I}(\rho;z) = \mathcal{H}(\rho) - \mathcal{H}(\rho|z) = \mathcal{H}(z) - \mathcal{H}(z|\rho)$
>
>
> By randomly selecting the latent variable $z$, we consider that $\mathcal{H}(z)$ is a constant depending on the number of $z$. Thus, we have,
>
> $\mathcal{H}(z|\rho) = \mathcal{H}(z) + \mathcal{H}(\rho|z) - \mathcal{H}(\rho)$
>
> $\\propto \\mathbb{E}_{(s,a,z)\\sim \\rho(s,a,z)}[-\\log{\\rho(s,a|z)}]$
>
> $- \\mathbb{E}_{(s,a)\\sim \\rho(s,a)}[-\\log \\rho(s,a)]$
>
> $=\mathbb{E}_{(s,a,z)\sim \rho(s,a,z)}[-\log \rho(s,a|z)] - \int -\rho(s,a)\log \rho(s,a) dsda$
>
> $=\mathbb{E}_{(s,a,z)\sim \rho(s,a,z)}[-\log \rho(s,a|z)] - \int -\rho(s,a,z)\log \rho(s,a) dsdadz$
>
> $=\mathbb{E}_{(s,a,z)\sim \rho(s,a,z)}[\log \rho(s,a)-\log \rho(s,a|z)]$
>
> Where, $\rho(s,a)=\int \rho(s,a|z)p(z) dz = \frac{1}{n}\sum_{k=1}^{n} p(s,a|z_k)$.
>
> Then,
>
> $\mathcal{H}(z|\rho) = \mathbb{E}_{(s,a,z)\sim \rho(s,a,z)}[-\log \rho(z|s,a)]$
>
> $= \mathcal{H}(z) + \mathcal{H}(\rho|z) - \mathcal{H}(\rho)$
>
> $\\propto \\mathbb{E}_{(s,a,z) \\sim \\rho(s,a,z)}$
>
> $\[\\frac{1}{n}\\sum_{k=1}^{n}\\rho(s,a|z_k)-\\log \\rho(s,a|z)\]$
>
> $\propto - \mathcal{D_{KL}}\left[\rho(s,a|z)||\frac{1}{n}\sum_{k=1}^{n} \rho(s,a|z_k)\right].$
>
> > Q5: There's a mismatch between Lemmas 2 and 3 in the appendix and the content presented in the main paper.
>
> A5: It's true that some of the symbols don't match the main paper, we'll fix that part, hopefully, this won't affect your understanding of this submission. For Lemma 3, we put some of the key formulas in the main paper which leads to an inconsistency.
>
> > Q6: The proof for (v) in Lemma 3 does not show the variance inequality: $V[X_{1:{N+1}}] \geq V[X_{1:{N}}]$.
>
> A6: Sorry about that. For some unknown reasons, our latest version was not uploaded correctly to the submission system. Our main body of this submission does not involve this result and we will remove this part in a subsequent version. Hopefully, this will not affect your understanding of our article.
>
> > Q7: Formatting Issue and Grammar.
>
> A7: Thanks for your advice. We will update these issues in the subsequent version.
>
> >L2: The computational overhead of the ensemble.
>
> A2: On the issue of computational cost, we use network with multi-head structure [4]. And with the use of recurrent training, our computational costs are highly reduced. To convinced that we calculate the training time over one epoch in HalfCheetah-v3 and compared with TD3, SAC and an ensemble method-SUNRISE. We list the results here. All the experiment are done in TITAN RTX 3060 with one single task mode.
>
> |Method | Epoch 1(s/Epoch)| Epoch 2 (s/Epoch)|Epoch 3 (s/Epoch)|Epoch 4 (s/Epoch)|Epoch 5 (s/Epoch)|
> |---|---|---|---|---|---|
> |TEEN (N=5) | 19.37| 18.35| 19.32| 17.49| 16.81|
> |TEEN (N=10) |18.35| 16.67|19.14 |19.09 |18.25 |
> |TEEN (N=15) |19.32 |18.89 |18.96 |19.84 |19.31 |
> |TD3|5.24 |4.68 |5.4 |5.63 |4.88 |
> |SAC| 11.74|8.51 |8.43 |8.44 |9.23 |
> |SUNRISE (N=5)|106.06 |105.9 |106.24 |102.29 |109.94 |
>
>
>
> [1] Kuznetsov A, Shvechikov P, Grishin A, et al. Controlling overestimation bias with truncated mixture of continuous distributional quantile critics[C]//International Conference on Machine Learning. PMLR, 2020: 5556-5566.
>
> [2] Fujimoto, Scott, Herke Hoof, and David Meger. "Addressing function approximation error in actor-critic methods." International conference on machine learning. PMLR, 2018.
>
> [3] Van Hasselt H, Guez A, Silver D. Deep reinforcement learning with double q-learning[C]//Proceedings of the AAAI conference on artificial intelligence. 2016, 30(1).
>
> [4] Osband I, Blundell C, Pritzel A, et al. Deep exploration via bootstrapped DQN[J]. Advances in neural information processing systems, 2016, 29.

---

> > ### Comment · Reviewer_vN15 · 2023-08-18
> >
> > I appreciate the thorough response from the authors. My concerns have been adequately addressed and I have increased the score accordingly.

---

> > > ### Author Response · Authors · 2023-08-18
> > >
> > > We are glad that our rebuttal could address your concerns. We appreciate your review and raising your score.

---

### Official Review · Reviewer_gTUk · 2023-07-04

**Soundness:** 3 good
**Presentation:** 2 fair
**Contribution:** 2 fair
**Rating:** 5
**Confidence:** 3

**Summary:**

This paper presents a new ensemble RL algorithm. The main motivation of this work is to increase the variation between sub-policies, which is measured by the distance between the corresponding state-action visitation distribution. It formulates the loss function by connecting the visitation distribution distance to the mutual information theory and introduces a discriminator function to measure the policy variation. It also proposes a few tricks to improve the exploration effectiveness of the proposed formulation. The method is experimented on a few Mujoco and Deepmind control suite tasks and compared against several off-policy RL methods and one ensemble RL method.

**Strengths:**

1. The proposed method is motivated by an intuitive idea that increasing the trajectory variation of the sub-policies may improve the exploration and performance of the ensemble policy.

2. The results show the proposed method consistently outperforms baselines across a wide span of tasks in different benchmarks.

3. The experimental analysis is helpful to understand the secret sauce behind the proposed method and the effect of different hyper-parameters.

**Weaknesses:**

1. The exposition of the paper need improvement (more in the next section)

2. The evaluated control problems are relatively simple and the approach needs to be evaluated on more complex tasks.

**Questions:**

1. Is the statement for the overestimation problem in DDPG properly demonstrated? Eq(4) (line 113) seems to be the formulation for discrete action space while this paper and DDPG focus on continuous action space. Please refer to [[9]](https://arxiv.org/pdf/1802.09477.pdf) for details.

2. It is unclear what the latent variable $z_k$ is at the beginning when it is first presented. Could be more clear if it just says $z_k$ is a categorical index or one-hot encoding of the sub-policy selection.

3. Line 146 mentions using KNN to measure Eq (8) but does the proposed approach indeed use it? If so, where is it applied?

4. Eq (9) is an important step to connect state-action visitation distribution distance to mutual information. Any references or derivation for Eq (9)?

5. The Recurrent Optimization trick (Line 169) is adopted to prevent sub-policy from exploration degradation. If I understand correctly, it corresponds to Line 18-19 in Algorithm 1 where only the sampled policy $\pi_k$ is updated. However, the gradient of the loss function (Eq (14)) should be independent among sub-policies. Could the authors explain how will the sub-policy $\pi_{\phi_k}$ change if we update all sub-policies instead of just $\pi_{\phi_k}$ in Algorithm 1 (Line 18-19).

6. **Algorithm 1** helps understand the method while there is no reference to Algorithm 1 throughout the paper.

7. How is the discriminator function $q$ trained? I could not find any details about it.

8. How does the presented algorithm perform on more complex control tasks? The current evaluated tasks are relatively simple that do not require a sophisticated amount of exploration of the policy. It would be more convincing if the approach can be benchmarked on Humanoid and Dextrous hand manipulation tasks.

9. The appendix is incomplete.

**Limitations:**

The limitations have been discussed above (i.e. it is unknown about the performance of the presented approach on complex tasks).

---

> ### Author Rebuttal · Authors · 2023-08-05
>
> We would like to extend our sincere gratitude for your insightful feedback on our submission. Your valuable comments have greatly contributed to enhancing the quality of our work. In response to your suggestions, we have carefully revised the manuscript to address the points you raised.
>
> > Q1: Is the statement for the overestimation problem in DDPG properly demonstrated? Eq(4) (line 113) seems to be the formulation for discrete action space while this paper and DDPG focus on continuous action space. Please refer to [9] for details.
>
> A1:  Overestimation bias occurs for similar reasons in both discrete and continuous environments, although the types of action spaces differ. We use a more explicit way of expressing overestimation bias in continuous action space and this expression is also recognized and used (eg. TQC [2]). Please check [1] and [2] for more information. We will give a more appropriate elaboration in a subsequent version.
>
>
>
> > Q2: It is unclear what the latent variable $z_k$ is at the beginning when it is first presented. Could be more clear if it just says $z_k$ is a categorical index or one-hot encoding of the sub-policy selection.
>
> A2: In our code, $z_k$ is a categorical index to select k-th sub-policy，(eg. $z=1,[\pi_1, \pi_2, \pi_3]=\\pi_1$). While the sub-policies can also be got by one-hot encoding as an input to the network. We don’t restrict how the sub-policy is obtained by $z_k$ because there are lots of ways to construct which is also a question worth exploring.
>
> > Q3: Line 146 mentions using KNN to measure Eq (8) but does the proposed approach indeed use it? If so, where is it applied?
>
> A3: We did not use this approach as we illustrate in lines 144-148. Estimating probability densities using KNN is computationally intensive and does not apply to continuous states.  In this paper, we avoided the problem of going to a direct solution for this difficult-to-estimate distribution by transforming Eq. using mutual information theory. The equations obtained in this way are precise and simpler to implement.
>
> > Q4: Any references or derivation for Eq (9)?
>
> A4: The Proof can be found in Appendix A. Where,
>
> $\mathcal{H}(\rho)=\mathbb{E}_{(s,a)\sim \rho}[-\log(\rho(s,a))]$
>
> $=\mathbb{E}_{(s,a,z)\sim \rho}\left[\log \frac{\rho(s,a|z)}{\rho(s,a)}-\log \rho(s,a|z)\right]$
>
> $= \mathbb{E}_{z}$
>
> $[\mathcal{D_{KL}}(\rho(s,a|z)||\rho(s,a))]+ \mathcal{H}(\rho|z)$
>
>
>
> > Q5: The Recurrent Optimization trick (Line 169) is adopted to prevent sub-policy from exploration degradation. If I understand correctly, it corresponds to Lines 18-19 in Algorithm 1 where only the sampled policy $π_k$ is updated. However, the gradient of the loss function (Eq (14)) should be independent among sub-policies. Could the authors explain how will the sub-policy $\pi_{\phi_{k}}$ change if we update all sub-policies instead of just π_{ϕ_{k}} in Algorithm 1 (Line 18-19).
>
> A5: if we update all sub-policies instead of just $\pi_{\phi_{k}}$ in Algorithm 1, We need a larger and gradually decaying coefficient alpha, and the results of this approach are much more brilliant. However, since the decay form of alpha is difficult to determine, thus, we used the Recurrent Optimization trick to avoid this problem which allows us to use a fixed coefficient $\alpha$ to get great behavior in most of the tasks.
>
>
> > Q6: Algorithm 1 helps understand the method while there is no reference to Algorithm 1 throughout the paper.
>
> A6: Thanks for your advice, we will add this reference at the appropriate place in the paper.
>
> > Q7: How is the discriminator function q trained? I could not find any details about it.
>
> A7: we train the discriminator parallelly with the policies, the update equation can be shown in Eq.(12) and we would add this part to our Alg.1 in the final version. Thanks.
>
> > Q8: How does the presented algorithm perform on more complex control tasks? The current evaluated tasks are relatively simple that do not require a sophisticated amount of exploration of the policy. It would be more convincing if the approach can be benchmarked on Humanoid and Dextrous hand manipulation tasks.
>
> A8: In this paper, we evaluate our algorithm in different exploration challenges from the MuJoCo control suite to the DeepMind Control suite. The experimental results show our algorithm improves both the performance and sample efficiency which reveals that these tasks do need more exploration and are still promising with a sophisticated amount of exploration.  But we'd love to challenge our algorithm in more complex benchmarks. Thus, we conduct experiments in the 376-dimensional Humanoid-v3, which is exceptionally difficult to solve with off-policy algorithms. We run an experiment in 5 million (5e6) time steps in 5 seeds. The results show that our algorithm exceeds the baseline algorithm consistently. We report the average returns with mean and variance of evaluation roll-outs here and the learning curves can be found in global rebuttal.
>
> | 5M Step | TEEN | TD3 |  SAC |
> | --- | :---: | :---: | :---: |
> | Humanoid-v3 | $8259.49\pm429.32$ | $6957.91\pm364.05$ |$7641.03\pm 261.63$|
>
> > Q9: The appendix is incomplete.
>
> A9: Sorry for that, we update some parts of the missing appendix in the general rebuttal.
>
> [1]Thrun, S. and Schwartz, A. Issues in using function approximation for reinforcement learning. In Proceedings of the 1993 Connectionist Models Summer School Hillsdale, NJ. Lawrence Erlbaum, 1993.
>
> [2] Kuznetsov A, Shvechikov P, Grishin A, et al. Controlling overestimation bias with truncated mixture of continuous distributional quantile critics[C]//International Conference on Machine Learning. PMLR, 2020: 5556-5566.

---

> > ### Comment · Reviewer_gTUk · 2023-08-15
> >
> > Thank you for the detailed response and new experiments with manipulation problems. I have some follow-up questions listed below:
> >
> > 1. The loss function in Algorithm 1 Line 19 is inconsistent with Eq. 14 in terms of the sign of the discriminator term. Is there anything I missed?
> >
> > 2. The answer A5 says we need to decay $\alpha$ if we would like to update all the policies. Why? and how does it connect to the motivations for the Recurrent Optimization (line 166-175)?
> >
> > Since the response addresses most of my concern, I would like to increase my rating if the questions above are properly resolved.

---

> > > ### Author Response · Authors · 2023-08-16
> > >
> > > We sincerely appreciate the feedback provided and are gratified to note that our rebuttal addressed a significant portion of your concerns. For the remaining questions, we clarify as follows.
> > >
> > > > Q1: The loss function in Algorithm 1 Line 19 is inconsistent with Eq. 14 in terms of the sign of the discriminator term.
> > >
> > > A1: We acknowledge your astute observation. Indeed, an oversight occurred in Algorithm 1, Line 19, where we inadvertently omitted a minus sign. This will be rectified in the forthcoming version. To elucidate, the correct formulation for Line 19 of Algorithm 1 should be
> > >
> > > $$\nabla_{\phi_k}\frac{1}{|B|}\sum_{(s,a,r,s')\in \mathcal{B}}(-\alpha\log q_{\zeta}(z_k|s,a)-Q_{\theta_k}(s,a)), a=\pi_{\phi_k}(s),$$
> > >
> > > This is subsequently followed by a clip operation, as described in Equation 15.
> > >
> > >
> > > > Q2: The answer A5 says we need to decay $\alpha$ if we would like to update all the policies. Why? and how does it connect to the motivations for the Recurrent Optimization (line 166-175)?
> > >
> > >
> > > A2: If we update all the policies simultaneously, we cannot perform recurrent optimization. So we need to find an approach to balance exploration and exploitation. As illustrated in lines 293-295 and the previous A5 response, a large $\alpha$ can enforce all the sub-policies to perform diverse explorations. Therefore, one generally potential method is the decay of $\alpha$. To encourage exploration, one needs to leverage a large $\alpha$ at the start of training to maintain the diversity of sub-policies. However, as training progresses, there emerges a necessity to strike a balance between exploration and exploitation. This entails a transition of $\alpha$ from a high to a more moderate value. Fine-tuning this balance can be challenging.
> > >
> > >
> > > However, we don't need to explicitly balance the exploration and exploitation if the recurrent optimization trick is utilized in the training. The essence of recurrent optimization lies in its ability to cyclically randomly select a sub-policy for exploration, accentuating its propensity for diverse exploration, while concurrently allowing other sub-policies to concentrate on exploitation. Thus, the recurrent optimization trick naturally balances the exploration and exploitation, rendering a constant $\alpha$ in our implementation wholly sufficient.
> > >
> > >
> > > If you have further questions, we would be happy to discuss them with you.

---

> > > > ### Comment · Reviewer_gTUk · 2023-08-17
> > > >
> > > > Thank you for your explanation. I've updated my score.

---

> > > > > ### Author Response · Authors · 2023-08-17
> > > > >
> > > > > Thank you for your updated score and insights!
> > > > >
> > > > > We value your reviews and welcome further discussions if there are any other concerns.

---

### Official Review · Reviewer_aBo4 · 2023-07-06

**Soundness:** 3 good
**Presentation:** 3 good
**Contribution:** 3 good
**Rating:** 6
**Confidence:** 3

**Summary:**

This paper presents an algorithm called trajectories-aware ensemble exploration (TEEN) for sample efficient ensemble reinforcement learning. The authors point out that the existing ensemble methods do not effectively address the required diversity in exploration, which TEEN is designed to tackle better. It achieves this by encouraging diverse behaviors through exploration of the state-action visit distribution measure space. Theoretical results are also presented to show how the design principles of TEEN could encourage diversity in exploration, while the experiment results demonstrate its empirical success in popular benchmark tasks.



**Strengths:**

This manuscript presents an interesting thesis that sample efficient ensemble reinforcement learning methods require diverse exploration in sub-policies. The authors discuss this topic in detail and systematically approach the proposed solution. I view the finding of the diversity requirement of sub-policies coupled with the provided theoretical analysis and the limited yet promising experimental results as the main strength of the work. I, therefore, believe if properly executed, this can be a valuable contribution to the community.



**Weaknesses:**

One of the main limitations of the work is its limited evaluations. The authors evaluate the proposed TEEN method to answer three main research questions. But I find their evaluations lacking the experimental rigor that is needed to fully appreciate the proposed method.

First, in RQ1 in evaluating the performance against baselines, the proposed method shows visible benefits in Mujoco tasks, which is promising. But my concerns are that all four tasks evaluated may contain similar exploration challenges. Given exploration is one of the key focuses of the work, I would like to see the performance benefits across different tasks that pose different exploration challenges. Perhaps this is what the authors are attempting to do with DeepMind control experiments, but despite the mention, I did not find such experiments in the paper or in the appendix (Appendix C is empty).

Regarding RQ2 and RQ3, my concerns are with the fact that it is only based on a single task (Ant-v3). For both questions, it is not significant enough to make claims based on a single task result. Furthermore, for RQ3, while I never found the results on the DeepMind control suite, the authors mention the best $\alpha$ is 0.02. This is a very low $\alpha$ and almost as having no trajectory-aware exploration. This further questions my concerns about how generalizable the proposed method is across different tasks. Clarification is appreciated.

Overall for experiments, authors use only TD3 as the learning algorithm. I would like to see if the observed performance gains are dependent on the choice of the learning algorithm. It is also fine if there is an observable reliance, but in that case, I would like to see the contributions rephrased accordingly.

I presume training the proposed model is challenging, specifically when it comes to finding the best ensemble size N and number of target values M (due to the trial and error approach in finding them, as in RQ2). It also appears that this training would require considerable computations. I would like to see a discussion of these limitations (and other limitations) of the method.

If the stated limitations are addressed, particularly related to the evaluation process, the proposed work can be of benefit to the community. Consequently, I would be inclined to reconsider my score if the authors can effectively rectify these issues.

**Questions:**

1. Shouldn’t the equation 11 be $\log(N) - \mathbb{E}_{s,a,z}[-\log(\rho(z|s,a))]$?
2. Also, shouldn’t the LHS and RHS of equation 11 be approximately equal and not equal?
3. Define $\mu$ and $\sigma$ in theorem 1.
4. While I did not check the proofs carefully, it appears the authors make simplification assumptions in the proof of theorem 1 that is not stated in the theorem (ex: zero-mean distribution). Please state them properly in the theorem.


**Limitations:**

Please discuss the limitations of the work, as pointed out earlier. This will help readers better appreciate the work.

---

> ### Author Rebuttal · Authors · 2023-08-07
>
> We would like to express our heartfelt appreciation for the invaluable feedback you provided on our submission. Your insightful comments and suggestions have been instrumental in refining our research. We clarify your concerns as follows.
>
> > W1: First, in RQ1 in evaluating the performance against baselines, the proposed method shows visible benefits in Mujoco tasks, which is promising. But my concerns are that all four tasks evaluated may contain similar exploration challenges. Given exploration is one of the key focuses of the work, I would like to see the performance benefits across different tasks that pose different exploration challenges. Perhaps this is what the authors are attempting to do with DeepMind control experiments, but despite the mention, I did not find such experiments in the paper or in the appendix (Appendix C is empty).
>
> A1: When evaluating our algorithm on MuJoCo tasks, we took that concern into account as well and that’s why we attempt to do with DeepMind control experiments as you infer. In Table 2, We report the average returns with mean and variance of evaluation roll-outs across all algorithms on DMControl Tasks with 5 other tasks. The results show that our algorithm far exceeds the baseline in a very small number of steps in most tasks. The learning curves for these tasks have been moved to Appendix.C.  Sorry for that, and check the general rebuttal for the missing part.
>
> > W2: Regarding RQ2 and RQ3, my concerns are with the fact that it is only based on a single task (Ant-v3). For both questions, it is not significant enough to make claims based on a single task result.
>
> A2: Regarding RQ2 and RQ3, the experiments for other environments have been moved to the appendix as we have mentioned in the main paper. We are sorry for the incomplete appendix, and check the general rebuttal for the missing part. And results are also consistent with our theoretical analysis.
>
> > W3: For the DeepMind control suite, the authors mention the best $\alpha$ is 0.02. ......  Clarification is appreciated.
>
> A3: We use a small $\alpha$ due to the small reward scale of the DeepMind control suite whereas in MuJoCo, the reward scale is 5 to 10 times of the DeepMind control suite. In the MuJoCo control suite, the accumulated reward exceeds 5k in Ant while in the DeepMind control suite, the max accumulated reward for all tasks is 1000. That’s why we use 0.2 for MuJoCo Control tasks while 0.02 for the DeepMind control suite encountering a small reward scale. And the ablation results also show that with higher alpha (0.5), our algorithm show better performance in HalfCheetah-v3 in which the reward scale is 2 times of Ant-v3.
>
> > W4: Overall for experiments, authors use only TD3 as the learning algorithm. I would like to see if the observed performance gains are dependent on the choice of the learning algorithm. It is also fine if there is an observable reliance, but in that case, I would like to see the contributions rephrased accordingly.
>
> A4: Yes, the gains are dependent on the choice of the learning algorithm. As we pointed out in Lemma 1 (Ensemble Sample Diversity Decomposition), where the ensemble diversity $\mathcal{H}(\rho)$ can be decomposed into two parts:
>
> $\mathcal{H}(\rho)=\mathbb{E}_{z}$
>
> $[\mathcal{D_{KL}}(\rho(s,a|z_k)||\rho(s,a))] + \mathcal{H}(\rho|z) $
>
> The first part is the state-action visit distribution discrepancy between the sub-policies and the ensemble policy induced by the KL-divergence measure and is our optimization target. While the second part implies the diversity of state-action visited by sub-policies which depends on which algorithm is used for the sub-policy. Thanks for your valuable suggestion, we’d include this in our contribution.
>
> > W5: I presume training the proposed model is challenging, specifically when it comes to finding the best ensemble size N and number of target values M (due to the trial and error approach in finding them, as in RQ2). It also appears that this training would require considerable computations. I would like to see a discussion of these limitations (and other limitations) of the method.
>
> A5: From the experimental results (More complete in the PDF of the general rebuttal.), Increasing the target value M will lead to underestimation and increasing the number of N can mitigate this. Thus, if we want to have a more accurate value function, a smaller M (no smaller than 2.) and a larger N is recommended.  However, a larger ensemble size can lead to considerable computations, and in most cases, N=10 is enough to make the value estimate more accurate. To alleviate the computational pressure, we adopt network with multi-head structures[1] and we compare our computational cost with TD3, SAC and an ensemble method-SUNRISE. The results can be found in personal rebuttal for R4. It takes about twice as long as what SAC takes for N=10 and one fifth of what SUNRISE takes.
>
> > Q1: Shouldn’t eq. 11 be $\log (N)−\mathbb{E}_{s,a,z}[−log(\rho(z|s,a))]$?
>
> A1: Sorry for this typo.
>
> > Q2: Also, shouldn’t the LHS and RHS of equation 11 be approximately equal and not equal?
>
> A2: We are trying to optimize the equation on the left, but the direct solution is not a good estimator of $p(s | z)$, so we transform it into a way to increase the lower bound of the equation while decreasing the gap between the lower bound and the true value of, see Eq. 12.
>
> > Q3: Define μ and σ in theorem 1.
>
> A3: Thanks for your advice, we consider $X_i\sim\mathcal{N}(\mu, \sigma)$, and we will add this definition in theorem 1 in the new version.
>
> > Q4: While I did not check the proofs carefully, it appears the authors make simplification assumptions in the proof of theorem 1 that is not stated in the theorem (ex: zero-mean distribution). Please state them properly in the theorem.
>
> A4: Thanks for your advice, we assume that $X_{i}\sim \mathcal{N}(\mu, \sigma)$, and we will add this assumption in theorem 1 in the new version.

---

> > ### Comment · Reviewer_aBo4 · 2023-08-17
> > **Thanks for the detailed response**
> >
> > The rebuttal is responsive on all major points, and the new experiments further confirm the findings (performance and ablation results) and highlight the limitations (computational cost). Therefore, I will raise my score accordingly.

---

> > > ### Author Response · Authors · 2023-08-17
> > >
> > > We appreciate your review and raising your score. Your thoughtful reviews strengthen our submissions. We are happy to continue discussions with you if there are any further questions.

---

### Official Review · Reviewer_C6dk · 2023-07-09

**Soundness:** 3 good
**Presentation:** 3 good
**Contribution:** 3 good
**Rating:** 5
**Confidence:** 4

**Summary:**

The paper presents an efficient ensemble learning strategy termed Trajectories-awarE Ensemble exploratioN (TEEN), which promotes exploration by diversifying the state-action visitation distribution of multiple sub-policies. The proposed method introduces a discrepancy measure that tries to maximize the difference in entropy of estimating the state-action visitation with and without the information of the policy (represented by the latent variable) and vice versa. Such a discrepancy is approximated by learning a discriminator that tries to differentiate between state-action pairs visited by different policies. And the log-likelihood of the discriminator is used as a regularizer for policy improvement along with maximizing the expected return of the policy. The authors employ additional strategies of updating a single policy at a time and clipping the discriminator log probabilities to ensure proper exploration. Finally, the estimation bias is controlled by calculating the target value as the minimum of the mean of the ensemble Q-values for each sub-policy. The choices are theoretically motivated and the algorithm is analyzed in simulation on Mujoco Control and DM Control Suite benchmarks.

**Strengths:**

1. TEEN provides new insights into using the measure of the state-action visitation distribution to promote exploration along with maximizing returns in policy optimization.

2. Each design choice made is properly justified: namely updating one sub-policy at a time and clipping the regularizer gradients.

**Weaknesses:**

1. There might be states not captured by the state-action visitation of one particular sub-policy (say $\pi_a$). In that case, the contribution of $\rho^{\pi_a}(s,a)$ to $\rho$(s,a) in eq (6) can be highly noisy and misinformative (garbage probability). Giving equal weightage to state-action visitation of every policy while calculating for the ensemble might be misleading.

2. It seems that the latent variable plays a key role in defining the algorithm and the discriminator. However, there is no discussion about the choice of $z_k$ and training of the discriminator in the paper or in Alg 1. How do you choose $z_k$ and how does the training take place of the discriminator? If done parallelly with the policies, the framework follows a min-max update rule, shown to be unstable in prior works like GAIL.

**Questions:**

1. How do you decide the actions while inference?
2. Can you clarify more about the weakness pt.2?
3. Section C is missing in the Supplementary.
4. I am unsure, but it seems that TEEN uses very high random exploration steps as mentioned in the supplementary. Do you use same for all the other algorithms?
5. If the regularization is already clipped, why is there a coefficient ($\alpha$) required? Can a comparison be shown between these two cases? Also, $\alpha=0$ means that there is only a change in Q-value computations as compared to TD3, hence there must be a result for all the environments with $\alpha=0$ and $\alpha=0.2$ to show a more convincing contribution of the regularization term. Currently, it is only shown for one env (Ant) and it is particularly not clear how much the regularization term is contributing to the exploration in addition to the ensemble of sub-policies.
6. The timesteps shown in the results are timesteps for individual sub-policy or the whole ensemble i.e. (1M/10 = 100k steps for single sub-policy)?

**Limitations:**

The authors have mentioned the limitation as the tuning of $\alpha$.

---

> ### Author Rebuttal · Authors · 2023-08-04
>
> We thank you for the your detailed feedback and insightful suggestions, please see the following for our response.
>
> > W1: There might be states not captured by the state-action visitation of one particular sub-policy(say $\pi_{a}$). In that case, the contribution of $\rho^{\pi_{a}}(s,a)$ to $\rho(s,a)$ in eq(6) can be highly noisy and misinformative (garbage probability). Giving equal weightage to state-action visitation of every policy while calculating for the ensemble might be misleading.
>
> A1: If a state is not captured by the state-action visitation of one particular sub-policy, this unexplored state gives equal large weightage to every policy and further enforces sub-policies to explore this unexplored state instead of misleading them through the reward function restricts policies to not explore aimlessly. Once the unexplored state is captured by the policy, there is no more misleading, as training proceeds.
>
> > W2: It seems that the latent variable plays a key role in defining the algorithm and the discriminator. However, there is no discussion about the choice of $z_k$ and training of the discriminator in the paper or in Alg 1. How do you choose z_k and how does the training take place of the discriminator? If done parallelly with the policies, the framework follows a min-max update rule, shown to be unstable in prior works like GAIL.
>
> A2: We sample $z_k$ from uniform distribution every 5k steps during recurrent training which can be shown in line 152 *“We fix p(z) to be uniform in this approach by randomly selecting one of the sub-policies to explore. We have $H(z) = −\frac{1}{N} \sum^{N}_{k=1} logp(z_k) ≈ logN$, which is a constant.”* Here, we fix $p(z)$ to be a prior distribution to guarantee that H(z) is a constant. Otherwise, there must be a more complicated form of the regular term If the distribution of z constitutes a posterior distribution through feedback. Uniform distribution means that the sub-policies are homogenous contributing equally to the ensemble exploration. As you understand, we train the discriminator parallelly with the policies, the update equation can be shown in Eq.(12) and we would add this part to our Alg.1 in the final version. The update of our algorithm is not a pure min-max update rule. TEEN updates the policy from both value function and the discriminator that you mentioned. The value function can help stabilize the gradient of policy and the exploration process further stabilizing our discriminator.
>
>
> > Q1: How to decide the actions while inference?
>
> A1: Check the response for W2.
>
> > Q2: Can you clarify more about the weakness pt.2?
>
> A2: Check the response for W2.
>
> > Q3: Section C is missing in the Supplementary.
>
> A3: Sorry about that, we upload some parts of Supplementary  C in the general rebuttal.
>
> > Q4: I am unsure, but it seems that TEEN uses very high random exploration steps as mentioned in the supplementary. Do you use same for all the other algorithms?
>
> A4: The random starting exploration time steps are set to be $2.5\times10^{4}$ in this submission. And this is a standard hyper-parameter [1], which keeps consistent for all the tested algorithms in our submission.
>
> > Q5: If the regularization is already clipped, why is there a coefficient (α) required? Can a comparison be shown between these two cases? Also, α=0 means that there is only a change in Q-value computations as compared to TD3, hence there must be a result for all the environments with $\alpha=0$ and $\alpha=0.2$ to show a more convincing contribution of the regularization term. Currently, it is only shown for one env (Ant) and it is particularly not clear how much the regularization term is contributing to the exploration in addition to the ensemble of sub-policies.
>
> A5: The reason we do regular term cropping is that when p in log p is small (e.g. 0.001), it will have a large gradient which will cause instability in the optimization process, whereas the coefficient alpha is designed to do a balance between exploration and exploitation, a higher coefficient alpha means that the agent is encouraged to explore the environment. See the PDF released in the general rebuttal for the evaluation when α=0.
>
> > Q6: The timesteps shown in the results are timesteps for individual sub-policy or the whole ensemble i.e. (1M/10 = 100k steps for single sub-policy)?
>
> A6: Yes. The timesteps shown are for the whole ensemble, which is 1M/10 = 100k steps for a single sub-policy if we use 10 sub-policies.
>
> [1] https://github.com/sfujim/TD3/blob/master/main.py

---

> > ### Author Response · Authors · 2023-08-19
> > **Further discussion would be welcome.**
> >
> > Dear Reviewer C6dk
> >
> > Thank you for your detailed review of our research. Your constructive comments are very helpful in helping us to improve our submissions. We truly value your feedback and have addressed your concerns in detail in our response. Have all your questions been answered? Since we have **less than** three days left for discussion, please let us know if there's anything else you'd like to discuss. If you're satisfied with our responses, we humbly hope you might consider giving the submission a **higher score** based on our response and your high comments on the Sound, Presentation, and Contribution.
> >
> > Best wishes,
> >
> > Authors.

---

### Author Rebuttal · Authors · 2023-08-09

We greatly appreciate the reviewer's thorough reading and detailed comments about our submission. Your constructive reviews strengthen our draft. And some major concerns might be due to potential misinterpretations of the paper and we would like to clarify them in the responses.
We updated Appendix C in this place.

**1. Performance evaluation on Deepmind control suite.**
In Figure 1, we present the performance curves on DMControl tasks. Our proposed algorithm TEEN also demonstrates superior performance on these tasks, which is consistent with our main paper. TEEN outperforms other baselines by a large margin on the fish-swim task.

**2. Ablation study.**
To understand the impact of hyper-parameters, we conducted experiments in various environments to show how the ensemble size N, the number of value functions M, and the weight $\alpha$ take effect. The results of the experiment are also in line with our expectations. The results can be found in Figure 2.

**3. Performance on a challenging task humanoid.**
To convince our evaluation, we conduct our algorithm in a challenging environment Humanoid-v3 in the MuJoCo suite, which is shown in Figure 3. The state dimension of Humaniod is 376 [^1], which is exceptionally difficult to solve. Our algorithm TEEN shows good performance in this task. We compared our algorithm  TEEN with sample-efficient algorithms TD3 and SAC. We follow the standard evaluation settings, carrying out experiments over five million (5e6) steps and running all baselines with 5 random seeds.

[^1] https://www.gymlibrary.dev/environments/mujoco/humanoid/

---

### Decision · Program_Chairs · 2023-09-21

**Decision:**

Accept (poster)

**Comment:**

This work present an approach that encourages exploration through ensemble policies. This results in more uniform state-action exploration, and ultimately to better performance.

All the reviewers agree that the paper present interesting insight into this important problem of RL.

Please address the feedback of the reviewers in your manuscript.